# A zero-agnostic model for copy number evolution in cancer

**Henri Schmidt, Palash Sashittal, Benjamin J. Raphael** *

Department of Computer Science, Princeton University, Princeton, New Jersey, United States of America

* braphael@princeton.edu

## Abstract

### Motivation

New low-coverage single-cell DNA sequencing technologies enable the measurement of copy number profiles from thousands of individual cells within tumors. From this data, one can infer the evolutionary history of the tumor by modeling transformations of the genome via copy number aberrations. Copy number aberrations alter multiple adjacent genomic loci, violating the standard phylogenetic assumption that loci evolve independently. Thus, specialized models to infer *copy number phylogenies* have been introduced. A widely used model is the *copy number transformation* (CNT) model in which a genome is represented by an integer vector and a copy number aberration is an event that either increases or decreases the number of copies of a contiguous segment of the genome. The CNT distance between a pair of copy number profiles is the minimum number of events required to transform one profile to another. While this distance can be computed efficiently, no efficient algorithm has been developed to find the most parsimonious phylogeny under the CNT model.

### Results

We introduce the *zero-agnostic copy number transformation* (ZCNT) model, a simplification of the CNT model that allows the amplification or deletion of regions with zero copies. We derive a closed form expression for the ZCNT distance between two copy number profiles and show that, unlike the CNT distance, the ZCNT distance forms a metric. We leverage the closed-form expression for the ZCNT distance and an alternative characterization of copy number profiles to derive polynomial time algorithms for two natural relaxations of the small parsimony problem on copy number profiles. While the alteration of zero copy number regions allowed under the ZCNT model is not biologically realistic, we show on both simulated and real datasets that the ZCNT distance is a close approximation to the CNT distance. Extending our polynomial time algorithm for the ZCNT small parsimony problem, we develop an algorithm, *Lazac*, for solving the large parsimony problem on copy number profiles. We demonstrate that *Lazac* outperforms existing methods for inferring copy number phylogenies on both simulated and real data.

**Data Availability Statement:** All software and scripts required to reproduce our results are available at https://github.com/raphael-group/lazac-copy-number. Archived data analyzed in the

manuscript is available on Zenodo (DOI: 10.5281/zenodo.8426943).

**Funding:** This work was supported by the National Cancer Institute (U24CA248453 and U24CA264027 to BJR). Authors PS and BJR received salary support from grants U24CA248453 and U24CA264027. HS received salary support from U24CA264027. The funders had no role in study design, data collection and analysis, decision to publish, or preparation of the manuscript.

**Competing interests:** The authors have declared that no competing interests exist.

## Author summary

Copy number aberrations are amplifications or deletions of large genomic regions, and occur frequently in cancer. However, reconstructing cancer evolution from copy number aberrations is challenging because unlike single-nucleotide mutations, copy number aberrations often overlap on the genome. Here, we introduce the *zero agnostic copy number transformation (ZCNT)* model to describe the accumulation of copy number aberrations in a genome. The ZCNT model accounts for some of the complexities of copy number evolution in a framework that enables scalable inference of evolutionary trees. We demonstrate the utility of the ZCNT model by efficiently reconstructing evolutionary trees for thousands of single cells from individual tumors. We anticipate that the ZCNT model will prove useful for future large-scale studies of tumor evolution at single-cell resolution.

## 1 Introduction

Tumor evolution is characterized by both small and large genomic alterations that alter the fitness of cancer cells [1]. *Copy number aberrations*, i.e. modifications to the number of copies of a genomic segment, are an important and frequent sub-class of such alterations that drive prognostic and metastatic outcomes [2]. Deriving the evolutionary history of copy number aberrations, herein referred to as *copy number phylogenies*, is thus important for understanding the emergence of primary tumors and the development of subpopulations of cells that evade treatment and/or metastasize to other anatomical sites.

Recent technological and computational improvements in single-cell sequencing have enabled the mapping of high resolution copy number profiles in single cells. For example, the high-throughput 10x Genomics Single-cell Copy Number Variation solution [3, 4] produces ultra-low coverage ($< 0.05\times$) whole genome sequencing data from $\approx 2000$ individual cells. Other recent technologies, including DLP/DLP+ [5–7] and ACT [8], produce similar data. Multiple computational methods [4, 9–13] have been introduced to infer high resolution *copy number profiles*, integer vectors that contain the number of copies of each genomic segment, from this type of data. Other recent methods can infer copy number profiles from thousands of cells or spatial locations from single-cell RNA sequencing (scRNA-seq) [14], scATAC-seq [15], or spatial transcriptomics data [16].

The increasing availability of technologies to measure genomic copy number in thousands of cell motivates development of methods to infer the cellular phylogenies from copy number profiles. However, there are multiple challenges in inferring phylogenies from copy number profiles. First, copy number aberrations are diverse, ranging from small duplications and deletions [17] to whole chromosome shattering and reconstruction events [18]. Second, a single copy number aberration can alter the number of copies of a large section of the genome *simultaneously*. This means that loci on the genome cannot be treated as independent phylogenetic characters, a widely-used assumption in phylogenetics [19–22]. Finally, the increasing size ($> 10, 000$ cells) and resolution ($< 5$Kb bins) of copy number profiles require increasingly scalable algorithms.

One widely used model of copy number evolution is the *copy number transformation* (CNT) model [23]. In the CNT model, a genome is represented as a vector of non-negative integers and copy number aberrations correspond to the increase or decrease of the entries in a contiguous *interval* of coordinates in the vector, explicitly modeling the non-independence of copy number amplifications and deletions. The *CNT distance* is the minimum number of copy number events needed to transformation one profile to another. The CNT distance is

computable in linear time [24] and has been used to define an evolutionary distance between profiles. Since the CNT distance is not symmetric, a variety of symmetrized CNT distances have also been used to construct copy number phylogenies using distance-based phylogenetic methods [23, 25–27]. Further, owing to its effectiveness, the CNT model has become the basis of a variety of distinct models [26–28] for copy number evolution.

While the CNT model is described by specific events—or mutations—there has been little work on constructing phylogenetic trees under the CNT model using the method of maximum parsimony. Even the small parsimony problem—where the topology of the tree is given and one aims to infer the ancestral profiles that minimizes the total number of copy number events on the tree—has no known efficient solution. For example, for the special case of a two leaf tree, the best algorithm for the CNT small parsimony problem [24] runs in $\mathcal{O}(nB^T)$ time where $B$ is the largest allowed copy number and $n$ is the number of loci [25]. Without an efficient algorithm for the small parsimony problem under the CNT model, one cannot hope to solve the large parsimony problem, where the topology of the tree is unknown.

We introduce a relaxation of the CNT model, called the *zero-agnostic copy number transformation* (ZCNT) model, that approximates the CNT model and has a number of desirable properties. Unlike the CNT model, the ZCNT model allows for the amplification of zero copy number regions. While such an operation is not biologically realistic, we show that this relaxation makes the ZCNT distance a metric, in contrast to the CNT distance. Moreover, we derive a closed form expression for the ZCNT distance between two profiles. We use this closed form expression as well as an alternative characterization of copy number profiles to solve two relaxations of the small parsimony problem in polynomial time and to derive a linear time, 2-approximation algorithm for the small parsimony problem. To our knowledge, this is the first attempt to solve the small parsimony problem for a segment-based (i.e. non-independent) model of copy number evolution. We then use our efficient algorithm for the (relaxed) small parsimony problem to design an algorithm, *Lazac* (Large-scale Analysis of Zero Agnostic Copy number), for inferring copy number phylogenies by solving the large parsimony problem. We show on simulated data that *Lazac* is > 100× faster than other phylogenetic methods and also more accurate in recovering the ground truth phylogeny. On single-cell whole-genome sequencing data from human breast and ovarian tumors, *Lazac* finds phylogenies that are more consistent with both copy number clones and single-nucleotide variants (SNVs).

## 2 Materials and methods

### 2.1 Copy number transformations

A *copy number profile* $p = [p_1, \ldots, p_m]$ is a vector of non-negative integers where $p_j \in \{0, \ldots, B\}$ is the the number of copies of locus $j$. Suppose we measure the copy number profile of $n$ cells of a tumor across $m$ loci in a single-cell DNA sequencing experiment. We encode the copy number profiles in a $n \times m$ *copy number matrix* $M = [M_{i,j}]$ where $M_{i,j} \in \{0, \ldots, B\}$ is the copy number of cell $i$ at locus $j$. The copy number profile $p$ of cell $i$ is then the $i^{\text{th}}$ row of this matrix, and is denoted $M_i$.

One of the most basic phylogenetic principles is that nearly perfect measurement of evolutionary distances enables exact recovery of the evolutionary history [29]. It is thus not surprising that many of the successful attempts at inferring copy number phylogenies focus on finding *good* methods to compute an evolutionary distance between a pair of copy number profiles. Early methods for computing evolutionary distances on copy number data [17, 30] employed simple measures of distance such as the Hamming, weighted Hamming, and $\ell_1$ distance between copy number profiles. However, these distances do not account for

dependencies between loci caused by long CNAs spanning contiguous segments of the genome, leading to inaccurate phylogenetic reconstruction [23, 27].

In this section, we describe and investigate the copy number transformation (CNT) model, one of the most well-known and successful evolutionary models for copy number evolution in cancer. The CNT model was originally introduced in MEDICC [23] and extended in subsequent studies [24–28]. Since the CNT model only allows intrachromosomal copy number events, it is sufficient to consider the case of a single chromosome, and thus for ease of exposition we will describe the model using a single chromosome.

The fundamental operation in the CNT model is a *copy number event* which increases or decreases (by one) the entries in a contiguous interval of a copy number profile, defined formally as follows.

**Definition 1** (Copy number event) *A copy number event $c_{s,t,b} : \mathbb{Z}_+^n \to \mathbb{Z}_+^n$ is a function that maps a copy number profile $p \in \mathbb{Z}_+^n$ to a profile $c_{s,t,b}(p)$ described by its entries as*

$$
c_{s,t,b}(p)_i = \begin{cases} p_i + b & \text{if } s \le i \le t \text{ and } p_i \ne 0, \\ p_i & \text{otherwise}, \end{cases}
$$

*where $s \le t$ and $b \in \{+1, -1\}$. We denote such a function as $c$ when clear by context.*

That is, an amplification (resp. deletion) increases (resp. decreases) the copy number of all *non-zero* entries in the interval between positions $s$ and $t$, or alternatively a copy number event *skips* the zero entries (Fig 1). Thus, once a locus is lost (i.e. $p_i = 0$), the locus cannot be regained or deleted further. A *copy number transformation (CNT) C* is the composition of multiple copy number events and we denote this function as $C = (c_1, \ldots, c_n)$ when $C(p) = c_n(\cdots(c_2(c_1(p))))$.

Several copy number problems have been previously studied to compute evolutionary distances under the CNT model. The first, and simplest, is the *copy number transformation problem*, originally introduced in [23], which defines a distance, $\sigma(u, v)$, between two copy number

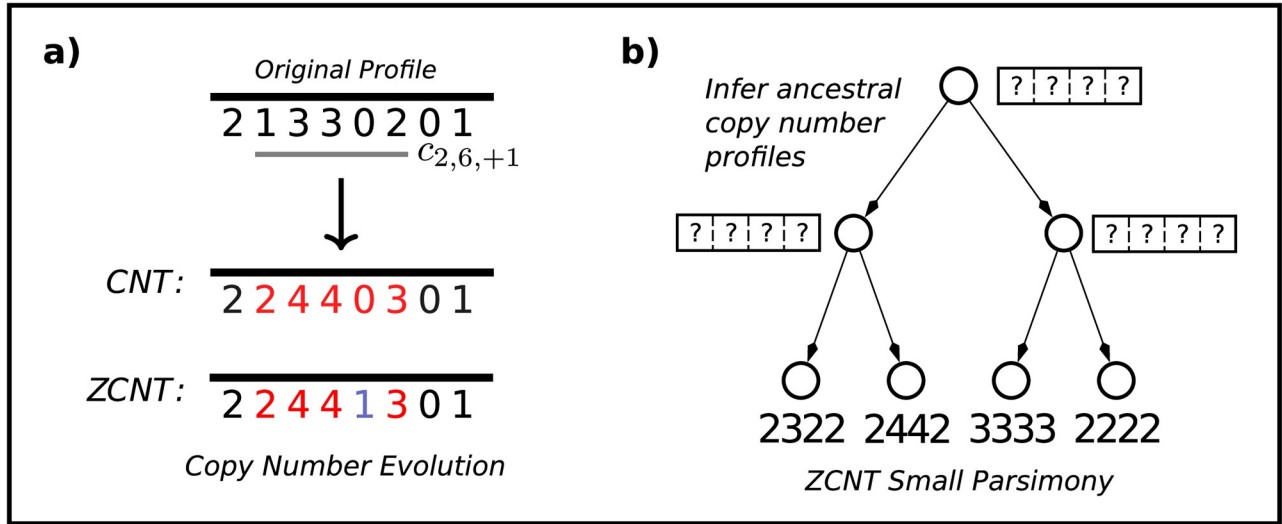

**Fig 1. (a)** The results of applying a copy number transformation $c_{2,6,+1}$ under both the CNT model and ZCNT model. Under the CNT model a zero cannot be increased via the amplification, but the zero-agnostic CNT model allows the zero to increase to one copy. **(b)** An instance of the ZCNT small parsimony problem: given a tree with copy number profiles labeling the leaves, the goal is to infer the ancestral copy number profiles that minimize the total ZCNT distance across all edges.

profiles. Put simply, the distance between two profiles is the length of the shortest copy number transformation needed to transform one profile to another.

**Definition 2** (Copy number transformation distance). *Given two copy number profiles u and v, the* copy number transformation distance *is*

$$\sigma(u, v) := \min_{C(u)=v} |C|,$$

*where $C = (c_1, \ldots, c_n)$ is a CNT. Alternatively, $\sigma(u, v) = \infty$ if no such transformation exists.*

[24, 31] show there is a (non-trivial) strongly linear time algorithm (i.e. time complexity $\mathcal{O}(|u| + |v|)$) for computing the CNT distance $\sigma(u, v)$. Unfortunately, the CNT distance $\sigma(u, v)$ is not symmetric (i.e. $\sigma(u, v) \neq \sigma(v, u)$), which makes it difficult to use in distance based phylogenetic methods such as neighbor joining [32].

In order to apply distance-based phylogenetic methods, multiple approaches to symmetrize the distance $\sigma(u, v)$ have been introduced. [24] use a mean correction replacing the asymmetric $\sigma(u, v)$ with a symmetric distance $\sigma'(u, v)$ defined as

$$\sigma'(u, v) = \frac{\sigma(u, v) + \sigma(v, u)}{2}.$$

Alternatively, several authors [23, 25, 27] define the distance between two profiles in terms of a closely related, median profile $w$. Specifically, the *median distance* between two profiles $u$ and $v$ is defined to be the smallest value of

$$\sigma(w, u) + \sigma(w, v)$$

over all profiles $w$. Computing this median distance is called the *copy number triplet problem* in [25]. Unfortunately, no efficient algorithm is known for the copy number triplet problem. The fastest algorithm uses $\mathcal{O}(mB^7)$ time and $\mathcal{O}(mB^4)$ space where $B$ is the maximum allowed copy number [25].

## 2.2 Small and large copy number parsimony

The small parsimony problem for copy number profiles is the following: given a tree $\mathcal{T}$ whose leaves are labeled by copy number profiles, infer ancestral copy number profiles that minimize the total dissimilarity between profiles across all edges (Fig 1). For evolutionary models in which each character evolves independently and has finitely many states (e.g. single nucleotide substitution models), the small parsimony problem is solved in polynomial time via Sankoff's algorithm, a dynamic programming algorithm [33]. Unfortunately, the CNT model presents two major challenges in solving the small parsimony problem. First, since copy number events affect multiple loci simultaneously, the loci cannot be analyzed independently, in contrast to most phylogenetic characters. Second, the space of possible copy number profiles is a priori unbounded, since the maximum copy number of a segment in a genome is unknown. Thus, it is not surprising that there is no published solution to the small parsimony problem for CNT dissimilarity, with the exception of the special case of two-leaf trees [25]. Here, we formalize both the CNT small parsimony problem and the corresponding large parsimony problem, the latter of which was previously described in [25].

A *copy number phylogeny* $(\mathcal{T}, \ell)$ is a rooted tree $\mathcal{T}$ and leaf labeling $\ell$. Let $V(\mathcal{T})$, $E(\mathcal{T})$, and $L(\mathcal{T})$ denote the vertices, edges, and leaves of $\mathcal{T}$, respectively. In our applications below, each leaf of $\mathcal{T}$ represents one of the $n$ cells (or bulk samples) from a tumor. An ancestral labeling $\hat{\ell}$ of a copy number phylogeny is a vertex labeling of $\mathcal{T}$ that agrees with $\ell$ on the leaves of $\mathcal{T}$, i.e. $\ell(v) = \hat{\ell}(v)$ when $v \in L(\mathcal{T})$. We say that $(\mathcal{T}, \ell)$ is a copy number phylogeny for copy number

matrix $M$ if $\mathcal{T}$ has $n$ leaves such that $\ell$ labels each leaf by a row of $M$. Formally, if $(\mathcal{T}, \ell)$ is a copy number phylogeny for a copy number matrix $M$, then there exists a cell assignment $\pi : [n] \to L(\mathcal{T})$ that assigns each cell $i$ to a leaf $v$ such that $\ell(v) = M_i$.

We define the cost $J(\mathcal{T}, \hat{\ell})$ of a vertex labeled, copy number phylogeny as the total number of copy number events required to explain the phylogeny:

$$J(\mathcal{T}, \hat{\ell}) := \sum_{(u,v) \in E(\mathcal{T})} \sigma(\hat{\ell}(u), \hat{\ell}(v)).$$

We now introduce the small parsimony problem [34] under the copy number transformation model.

**Problem 1** (CNT small parsimony). *Given a copy number phylogeny $(\mathcal{T}, \ell)$ find an ancestral labeling $\hat{\ell} : V(\mathcal{T}) \to \{0, \ldots, B\}^m$ such that (i) $\hat{\ell}(v) = \ell(v)$ for all leaves $v \in L(\mathcal{T})$ and (ii) $J(\mathcal{T}, \hat{\ell})$ is minimized.*

The *parsimony score* is defined as the cost $J(\mathcal{T}, \hat{\ell})$ of the solution $\hat{\ell}$ to the CNT small parsimony problem. To the best of our knowledge the CNT small parsimony problem (Problem 1) has not been analyzed in the literature. We believe this is due to the difficulty of solving the CNT small parsimony problem. That is, for even a special case of two-leaf trees, referred to as the *copy number triplet* problem [25], no strongly polynomial time algorithm is known (Section 2.1).

The CNT large parsimony problem defined in [25] aims to find a vertex labeled, copy number phylogeny $(\hat{\mathcal{T}}, \ell)$ for a matrix $M$ with minimum cost such that the root of $\mathcal{T}$ is labeled by the normal, diploid copy number profile.

**Problem 2** (CNT large parsimony). *Given copy number matrix $M$, find a copy number phylogeny $(\mathcal{T}, \ell)$ for $M$ and ancestral labeling $\hat{\ell}$ such that $J(\mathcal{T}, \hat{\ell})$ is minimized and $\ell(r) = (2, \ldots, 2)$ for the root $r$ of $\mathcal{T}$.*

Unsurprisingly, [25] showed that the above large parsimony problem (Problem 2) is NP-hard. They also formulated an integer linear program (ILP) to solve the problem exactly. However, this ILP consists of $O(n^2 m + nm \log B)$ variables and does not scale to the size of current real data sets with thousands of cells.

## 2.3 The zero-agnostic CNT model

The copy number transformation (CNT) model imposes the constraint that once a locus is lost (has zero copy number), the locus remains with zero copies for all time. While this constraint is biologically realistic, the constraint also makes the inference problems—including the CNT small (and large) parsimony problems—computationally hard to solve. Here, we show that *relaxing* the constraint that copy number events do not alter zero entries leads to a simpler model with favourable mathematical properties. We call this the *zero-agnostic* copy number model (Fig 1) to indicate that the model allows the amplification and deletion of loci with zero copies. Formally, we define a *zero-agnostic copy number event* as follows.

**Definition 3** (Zero-agnostic copy number event). *A zero-agnostic copy number event $c_{s,t,b} : \mathbb{Z}^n \to \mathbb{Z}^n$ is a function that maps a profile $p \in \mathbb{Z}^n$ to a profile written $c_{s,t,b}(p)$ described by its entries as*

$$c_{s,t,b}(p)_i = \begin{cases} p_i + b, & \text{if } s \leq i \leq t, \\ p_i & \text{otherwise,} \end{cases}$$

*where $s \leq t$ and $b \in \{+1, -1\}$. We denote such a function as $c$ when clear by context.*

Thus, a zero-agnostic copy number event either increases or decreases the number of copies of all loci in the interval $(s, t)$ regardless of whether the loci have zero copies. Analogous to our definition of a copy number transformation, we define a *zero-agnostic copy number transformation C* as the composition of multiple zero-agnostic copy number events and denote this function as $C = (c_1, \ldots, c_n)$ where $C(p) = c_n(\cdots(c_2(c_1(p))))$. Note that our formulation of a zero-agnostic copy number transformation (ZCNT) allows for the number of copies of a locus to decrease below zero, one can show that given two profiles with non-negative entries, it is always possible to find a minimum length zero-agnostic copy number transformation such that no intermediate profile has negative entries. See Section 2.3.2 below.

Due to space constraints, we do not include all proofs in the main text. Any proof not present in the main text can be found in Sections B and C in S1 Text.

**2.3.1 Delta profiles.** We simplify our analysis of zero-agnostic copy number events by examining their effect on the differences between the copy number of adjacent loci. In particular, while a zero-agnostic copy number event $c_{s,t,b}$ increments (or decrements) all entries $p_i$ where $i \in \{s, \ldots, t\}$, $c_{s,t,b}$ only alters two *differences* between adjacent loci, namely the difference $p_s - p_{s-1}$ and the difference $p_{t+1} - p_t$. To formalize this idea, we first define the *delta profile*, a vector obtained by taking the differences in copy number between adjacent loci.

**Definition 4**. *A* delta profile *is any vector* $q \in \mathbb{Z}^n$ *that satisfies the* balancing condition:

$$\sum_{i=1}^{n} q_i = 0. \tag{1}$$

*Or equivalently,* $\sum_{q_i>0} |q_i| = \sum_{q_i<0} |q_i|$. *We denote the set of delta profiles in* $\mathbb{Z}^n$ *as* $\mathcal{D}_n$.

The above definition provides us with a convenient (and useful) description of the image of the following difference transformation, which we call the *delta map*.

**Definition 5**. *The* delta map $\Delta : \mathbb{Z}^n \to \mathcal{D}_{n+1}$ *maps a copy number profile p to a delta profile* $\Delta(p)$ *by taking the differences in the copy number of adjacent loci after appending normal, diploid copy number loci to both ends of p. Specifically,*

$$\Delta(p)_1 = p_1 - 2, \ \Delta(p)_i = p_i - p_{i-1}, \ and \ \Delta(p)_{n+1} = 2 - p_n$$

*where the constant* 2 *represents a normal, diploid copy number.*

A basic property of the delta map $\Delta$ is that it is invertible.

**Proposition 1**. *The delta map* $\Delta : \mathbb{Z}^n \to \mathcal{D}_{n+1}$ *is invertible.*

Since $\Delta$ is one-to-one and onto with respect to $\mathcal{D}_{n+1}$, each delta profile $q$ then corresponds to a unique copy number profile $p = \Delta^{-1}(q)$.

Interestingly, a copy number event $c_{s,t,b}$ applied to a copy number profile $p$ only affects two entries of the delta profile $\Delta(p)$, meaning that loci of the corresponding delta profile are (nearly) independent. We formalize this in the following definition of a *delta event*.

**Definition 6** (Delta event). *A* delta event $\delta_{s,t,b} : \mathcal{D}_n \to \mathcal{D}_n$ *is a function that maps a delta profile* $q \in \mathcal{D}_n$ *to a delta profile* $\delta_{s,t,b}(q)$ *described by its entries as*

$$\delta_{s,t,b}(q)_i = \begin{cases} q_i + b & if \ i = s \\ q_i - b & if \ i = t + 1 \\ q_i & otherwise, \end{cases}$$

*where* $s \leq t$ *and* $b \in \{+1, -1\}$. *We denote such a function as* $\delta$ *when clear by context.*

A *delta transformation* $D = (\delta_1, \ldots, \delta_n)$ is the composition of multiple delta events, where $D(q) = \delta_n(\cdots(\delta_2(\delta_1(q))))$. We now state the connection between delta events and zero-agnostic copy number (ZCNT) events in the following theorem and corollary.

**Theorem 1**. *Let $c_{s,t,b}$ be a zero-agnostic copy number event and $\delta_{s,t,b}$ be a delta event. Then,*

$$p' = c_{s,t,b}(p) \quad if \ and \ only \ if \quad \Delta(p') = \delta_{s,t,b}(\Delta(p)).$$

**Corollary 1**. *Let $C = (c_{s_1,t_1,b_1}, \ldots, c_{s_n,t_n,b_n})$ be a zero-agnostic copy number transformation and $D = (\delta_{s_1,t_1,b_1}, \ldots, \delta_{s_n,t_n,b_n})$ be the corresponding delta transformation. Then,*

$$p' = C(p) \quad if \ and \ only \ if \quad \Delta(p') = D(\Delta(p))$$

*Proof*. The corollary follows by induction on $|C|$ and repeated application of (Theorem 1).

**2.3.2 Computing the ZCNT distance.** Let $d(p, p')$ be the minimum number of *zero-agnostic* copy number events needed to transform the copy number profile $p$ to $p'$. In this section we derive a closed form expression for $d(p, p')$.

We begin by noting that $d(p, p')$ is equal to the minimum number $d'(\Delta(p), \Delta(p'))$ of delta events needed to transform delta profile $\Delta(p)$ to $\Delta(p')$. This follows from the equivalence between the copy number transformations and the corresponding delta transformation (Corollary 1). Thus, it suffices to only consider delta profiles and delta events; for the rest of the section all profiles $q$ and $q'$ are delta profiles unless otherwise specified.

We start by observing two basic facts: delta transformations are commutative and $d'(q, q')$ forms a metric.

**Proposition 2**. *A delta transformation $D = (\delta_1, \ldots, \delta_n)$ is commutative. That is, the application of $D$ to a profile is identical to the application of $D_\sigma = (\delta_{\sigma_1}, \ldots, \delta_{\sigma_n})$ where $\sigma$ is any permutation of $\{1, \ldots, n\}$.*

**Proposition 3**. *$d'(q, q')$ is a distance metric. Further,*

$$d'(q, q') = d'(q - q', 0) = d'(q' - q, 0).$$

Note that this also implies that zero-agnostic copy number transformations are commutative and that $d(\cdot, \cdot)$ is a distance metric. To see this, let $C = (c_{s_1,t_1,b_1}, \ldots, c_{s_n,t_n,b_n})$ be a zero-agnostic copy number transformation and $D = (\delta_{s_1,t_1,b_1}, \ldots, \delta_{s_n,t_n,b_n})$ be the corresponding delta transformation, then for any pair of copy number profiles $p, p' \in \mathbb{Z}^n$ and permutation $\sigma$,

$$C(p) = p' \Leftrightarrow D(\Delta(p)) = \Delta(p') \Leftrightarrow D_\sigma(\Delta(p)) = \Delta(p') \Leftrightarrow C_\sigma(p) = p',$$

where the first equivalence follows from Corollary 1, the second from Proposition 2, and the third from Corollary 1. This implies that $C(p) = C_\sigma(p)$, which proves that a zero-agnostic copy number transformation is commutative. To see that $d(\cdot, \cdot)$ is a distance metric, it suffices to observe that $d(p, p') = d'(\Delta(p), \Delta(p'))$ implies symmetry and reflexivity. The triangle inequality is satisfied since the composition of a zero-agnostic copy number transformation from $p$ to $r$ and $r$ to $p'$ yields a copy number transformation from $p$ to $p'$.

As a corollary to the commutativity of zero-agnostic copy number transformations, given any zero-agnostic copy number transformation $C$, one can re-order the events such that amplifications always occur first. After this reordering of events, no intermediate profile has negative entries as long as both the initial and resultant profile have no negative entries. Thus, it is always possible to find a minimum length zero-agnostic copy number transformation such that no intermediate profile has negative entries.

From our characterization of delta profiles, we derive our expression for the distance between delta profiles.

**Theorem 2**. *For all delta profiles $q$ and $q'$ in $\mathcal{D}_n$, we have $d'(q, 0) = \frac{1}{2}\|q\|_1$. Thus, $d'(q, q') = \frac{1}{2}\|q - q'\|_1$.*

*Proof.* Since each delta event decreases the total magnitude of $\|q\|_1$ by at most two, to transform $q$ to the 0 profile requires at least $\frac{1}{2}\|q\|_1$ events.

We prove the other direction by induction on $\sum_{q_i>0}|q_i|$. Clearly, if the sum is zero, the claim holds. Otherwise, by the balancing condition (1), we can choose $\delta$ to be any delta event that decrements $i \in \{i : q_i > 0\}$ and increments $j \in \{j : q_i < 0\}$. Applying $\delta$ to $q$ results in a delta profile $\delta(q)$ such that $\|\delta(q)\|_1 = \|q\|_1 - 2$. Invoking the induction hypothesis then yields a sequence of $\frac{1}{2}\|q\|$ delta events to transform $q$ to the 0 profile.

The second statement follows from Proposition 3.

As a corollary to the above theorem and the equivalence between zero-agnostic copy number transformations and delta transformations (Corollary 1), we have our closed form expression for the ZCNT distance between copy number profiles.

**Corollary 2**. *For all copy number profiles $p$ and $p'$ in $\mathbb{Z}^n$,*

$$d(p, p') = d'(\Delta(p), \Delta(p')) = \frac{1}{2}\|\Delta(p) - \Delta(p')\|_1.$$

Further, as a corollary to the fact that $d(p, p')$ is a distance metric, the following median distance is trivially computed in linear time:

**Corollary 3**. *Given two copy number profiles $p$ and $p'$ in $\mathbb{Z}^n$, both $p$ and $p'$ minimize the median distance $d(r, p) + d(r, p')$ over all choices of copy number profiles $r$. Thus,*

$$\min_{r \in \mathbb{Z}^n} \{d(r, p) + d(r, p')\} = d(p, p').$$

## 2.4 ZCNT small parsimony

We show below that the special form of the ZCNT model enables us to solve three variants and special cases of the small parsimony problem polynomial time. First, using the equivalence between copy number profiles and delta profiles described above, we formulate the small parsimony problem (Problem 1) using the ZCNT model as follows, where we drop the constant factor of 1/2 for ease of exposition.

**Problem 3** (ZCNT Small Parsimony). *Given a copy number matrix $M$, a tree $\mathcal{T}$ and an assignment $\pi$ of cells to leaves, find a vertex labeling $\ell : V(\mathcal{T}) \rightarrow \mathbb{Z}^m$ minimizing the cost $J(\mathcal{T}, \ell)$, or equivalently the sum*

$$\sum_{(u,v) \in E(\mathcal{T})} \|\ell(u) - \ell(v)\|_1,$$

*such that the following two conditions are satisfied:*

*i. $\ell(\pi(i)) = \Delta(M_i)$ for all cells $i \in [n]$,*

*ii. $l(u)$ satisfies the balancing condition (1) for all vertices $u \in V(\mathcal{T})$.*

To solve the above problem, we recall the general form of the Sankoff-Rousseau recurrence [33, 35] for solving the small parsimony problem. Let $c(\mathcal{T}; x)$ be the cost of the optimal labeling $\hat{\ell}$ of $V(\mathcal{T})$ that satisfies condition (i) of Problem 3 and has label $x$ for the root. Let $\mathcal{T}_w$ denote the sub-tree rooted at $w$ and suppose that $w$ has children $u$ and $v$. Then, by condition (ii), and the requirement that the ancestral labeling lies in $\mathbb{Z}^m$, we have the following

recurrence relation [35]:

$$c(\mathcal{T}_w; x) = \min_{y,z \in \mathcal{D}_n} \{\|x - y\|_1 + \|x - z\|_1 + c(\mathcal{T}_u; y) + c(\mathcal{T}_v; z)\}. \tag{2}$$

This recurrence has several difficulties. First, $y$ and $z$ are unbounded and can take on any value in $\mathbb{Z}^m$. Thus, it is impossible to store a dynamic programming table for $c(T_w; x)$ without imposing bounds on the maximum copy number. Further, even when the entries are constrained to a bounded interval $\{0, \ldots, B\}^m$, the dynamic programming table has size $(B + 1)^m$, exponentially large. Second, because of the balancing conditions (1), $\sum_{i=1}^n y_i = \sum_{i=1}^n z_i = 0$, one cannot analyze the loci independently.

**2.4.1 Two polynomial time relaxations.**   Despite these challenges, the recurrence (2) is a substantial improvement over the analogous recurrence under the CNT model. In fact, if we remove *either the balancing* (1) *or the integrality condition*, we can solve this recurrence in (resp. strong or weak) polynomial time. These relaxations not only provide us with lower bounds for the value of the true solution that can be computed efficiently, but these lower bounds are empirically quite tight (Section 3.4).

**Theorem 3** *If the balancing condition* (1) *is dropped, the ZCNT small parsimony problem can be solved in $\mathcal{O}(mn)$ time. If the integrality condition is dropped, the ZCNT small parsimony problem can be solved in (weakly) polynomial time using a linear program with $\mathcal{O}(mn)$ variables and constraints.*

Both of these facts derive from our closed form expression for the ZCNT distance between two copy number profiles in terms of the $\ell_1$ norm. We sketch the ideas here, and refer to Sections B.1 and B.2 in S1 Text for proofs of these claims.

For the first case when we drop the balancing condition, we can analyze the loci independently as there is no constraint on the entries of the ancestral profiles. Then, it suffices to observe that since the distance corresponds to the absolute difference, the function $c(\mathcal{T}_w; x)$ has a nice structure and we do not have to store an infinitely large dynamic programming table. When the integrality is removed, then, since both (i) and (ii) are *linear* constraints on the profiles and because $\ell_1$ norm minimization can be written as a linear program (LP), there is an LP formulation of the ZCNT small parsimony problem. As it is well known we can solve LPs in (weakly) polynomial time, this concludes the second case.

**2.4.2 A relabeling strategy and approximation algorithm.**   Since solutions to the relaxed ZCNT small parsimony problem do not necessarily yield solutions to the original problem, e.g. by not satisfying the balancing condition, it is natural to ask if it is possible to "fix" such solutions. Here, we show that given *any* ancestral labeling of a tree $\mathcal{T}$, we can perform local fixes that do not substantially increase the small parsimony objective and ensure that the resultant labeling satisfies the balancing condition. Then, we will show that these local fixes lead to a linear time, 2-approximation algorithm for the ZCNT small parsimony problem. While the results here are stated (and proven) only for *binary* trees, these results are without loss of generality as polytomies can be (arbitrarily) resolved to create binary trees without increasing the ZCNT small parsimony score. We sketch the ideas here and refer to Section B.3 in S1 Text for proofs of these claims.

Given a labeling $\ell : V(\mathcal{T}) \to \mathbb{Z}^m$ of $\mathcal{T}$, the *discrepancy* of a vector $l(u)$ is defined as

$$\text{disc}(l(u)) := \sum_{j=1}^m \ell(u)_j,$$

and is also called the discrepancy of the vertex $u$. Interestingly, the vertex discrepancy provides

us with a bound on the cost to relabel $\ell$ to ensure it satisfies the balancing condition (1), as stated in the following corollary.

**Corollary 4.** *Let* $\ell : V(\mathcal{T}) \to \mathbb{Z}^m$ *be any labeling of a binary tree* $\mathcal{T}$ *such that* $\ell(\pi(i)) = \Delta(M_i)$ *for all cells* $i \in [n]$. *Then, we can construct a new labeling* $\ell'$ *satisfying the balancing condition* (1) *such that*:

$$J(\mathcal{T}, \ell') \leq J(\mathcal{T}, \ell) + \sum_{u \in V(\mathcal{T})} |disc(\ell(u))|.$$

*Further, we can compute the labeling* $\ell'$ *in* $\mathcal{O}(mn)$ *time.*

Said another way, the above corollary provides us with an approximation algorithm for the ZCNT small parsimony problem where the approximation error is bounded by the total discrepancy. In fact, this general strategy gives us a linear time, 2-approximation algorithm. To see this, let $\ell$ be a solution to the ZCNT small parsimony problem when the balancing condition (1) is dropped. Then, the cost $J(\mathcal{T}, \ell)$ is at least as large as the total discrepancy, as stated in the following lemma.

**Lemma 1.** *Let* $\ell : V(\mathcal{T}) \to \mathbb{Z}^m$ *be any labeling of a binary tree* $\mathcal{T}$ *such that* $\ell(\pi(i)) = \Delta(M_i)$ *for all cells* $i \in [n]$. *Then*

$$J(\mathcal{T}, \ell) \geq \sum_{u \in V(\mathcal{T})} |disc(\ell(u))|.$$

Then, relabel $\ell$ using the above strategy to obtain $\ell'$. Since $J(\mathcal{T}, \ell)$ is a lower bound of the true cost and since $J(\mathcal{T}', \ell)$ is at most twice $J(\mathcal{T}, \ell)$, the cost of $\ell'$ is at most twice the true cost. Further, since we can compute both the $\ell$ and $\ell'$ in linear time, this leads to the following theorem.

**Theorem 4.** *For binary trees, the ZCNT small parsimony problem can be solved to within a constant factor of 2 in* $\mathcal{O}(mn)$ *time.*

**2.4.3 A special case: The multiple median problem.** We conclude this section with an investigation into a special case where the input tree has a single internal node, also known as the (multiple) median problem [25]. More formally, let $p_1, \ldots, p_n$ be a set of copy number profiles, the *ZCNT median problem* is to compute the the profile $r$ minimizing the total distance:

$$d(r, p_1) + \ldots + d(r, p_n) = d'(\Delta(r), \Delta(p_1)) + \ldots + d'(\Delta(r), \Delta(p_n)).$$

We can solve this with an elegant dynamic program. Define $A[j, k]$ as the minimum of

$$d'(r[1\ldots j], \Delta(p_1)[1\ldots j]) + \cdots + d(r[1\ldots j], \Delta(p_n)[1\ldots j])$$

over all $r$ where $r[1\ldots j]$ is the $j^{\text{th}}$ prefix of $r$ and $r[1\ldots j]$ has discrepancy $k$. Then, clearly $A[m, 0]$ is the score of the optimal solution to the ZCNT median problem, since if $r$ has discrepancy zero it satisfies the balancing condition (1). It then suffices to show that we can efficiently compute $A[j, k]$.

To compute $A[j, k]$, first notice there is always an optimal solution such that $r_j \in \{-B, \ldots, B\}$ for all $j \in [m]$ where $B$ is the maximum copy number in the input. Then, the vertex discrepancy $k$ is in $\{-Bm, \ldots, Bm\}$, and we can represent $A[j, k]$ as an $m \times Bm$ matrix. Matrix $A$ then satisfies the recurrence

$$A[j, k] = \min_{b \in \{-B, \ldots, B\}} \left\{ A[j-1, k-b] + \sum_{i=1}^{n} |b - \Delta(p_i)_j| \right\},$$

since if $r[1\ldots j]$ ends in $b$ and has discrepancy $k$, the prefix $r[1\ldots j-1]$ has discrepancy $k - b$.

For the boundary conditions, we set $A[i, k] = 0$. As it takes $2Bn$ computations to fill in each $A[j, k]$, it takes $\mathcal{O}(nm^2B^2)$ total time to fill in $A$. Reading off a solution takes the same time. Thus, we have solved the ZCNT median problem in $\mathcal{O}(nm^2B^2)$ time.

## 2.5 *Lazac* algorithm for ZCNT large parsimony

We develop a tree-search algorithm, *Lazac*, to find approximate solutions to the ZCNT large parsimony problem (Problem 2). Our procedure searches the space of copy number trees for a given copy number matrix $C$ using sub-tree interchange operations [36] and relies heavily on the efficient algorithm we developed for the small parsimony problem (Problem 3) when the balancing condition (1) is dropped. The procedure is similar to the tree search procedure we developed for lineage tracing data [37] based off IQ-TREE [36]. Complete details on our tree search procedure are in Section A in S1 Text *Lazac* is implemented in C++17 and is freely available at: `github.com/raphael-group/lazac-copy-number`.

## 3 Results

### 3.1 Comparison of copy number distances and phylogenies on prostate cancer data

We first investigated the differences between the CNT and ZCNT distances on copy number profiles inferred from bulk whole-genome sequencing data from ten metastatic prostate cancer patients [38]. We analyzed the copy number profiles for these patients published in [27]. For each pair of copy number profiles from distinct samples (e.g. anatomical sites) from the same patient, we computed the CNT distance $d_{\text{CNT}}$ and ZNCT distance $d_{\text{ZCNT}}$. We found that for all ten patients, the median relative difference $|d_{\text{CNT}}/d_{\text{ZCNT}} - 1|$ over all pairs of samples was less than 5%—and for many patients the relative difference was even smaller (S5 and S6 Figs).

### 3.2 Evaluation on simulated data

We compared *Lazac* to several state-of-the-art methods for inferring copy number phylogenies—namely MEDICC2 [27], MEDALT [31], Sitka [39], and WCND [26]—on simulated data.

*Lazac* inferred the most accurate phylogenetic trees across varying number of cells ($n = 100$, 150, 200, 250, 600) and loci ($l = 1000, 2000, 3000, 4000$). In particular, we found that on all but one parameter setting, *Lazac* had the lowest median RF (Fig 2) and Quartet distance (Section A.5 in S1 Text) on simulated instances (S1 Fig). Further, on large phylogenies containing $n = 600$ cells, *Lazac* showed an even larger improvement in median RF (Fig 2) and Quartet distance (Section A.5 in S1 Text) over other methods, owing to its scalability. In terms of speed, *Lazac* was the fastest method on every instance, taking less than $\sim 250$ seconds to run on the largest simulated dataset containing 600 cells. Further, it was $\sim 100$ times faster than the other top performing methods Sitka and MEDICC2 (Fig 2B).

As a further evaluation of the differences between the ZCNT and CNT distances, we compared the trees obtained using distance-based phylogenetic methods with the ZCNT and CNT distances. Specifically, we compared the performance of applying neighbor joining on the ZCNT distances, referred to as *Lazac-NJ*, to three distance-based methods for reconstructing copy number phylognies: MEDICC2 [27], WCND [26], and MEDALT [31] on simulated data. MEDICC2 and WCND compute distances based on extensions of the CNT model and then apply neighbor joining to infer phylogenies. As such, they allow for a natural benchmark with which to compare our simpler, ZCNT distance. *Lazac-NJ* had nearly identical (within 1%) median RF and Quartet distance compared to other distance based methods (S2 and S3 Figs).

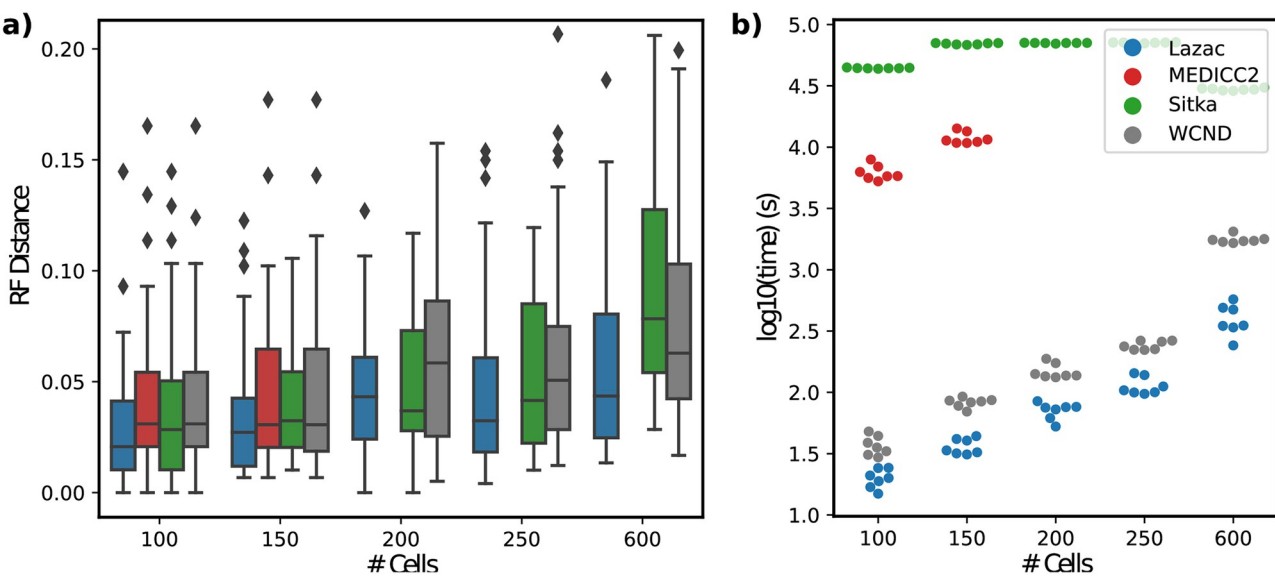

**Fig 2. (a)** Comparison of reconstruction accuracy (RF distance) on simulated data for several state-of-the art methods for copy number tree reconstruction with varying number of cells $n$ = 100, 150, 200, 250, 600 across four sets of loci $l$ = 1000, 2000, 3000, 4000 and seven random seeds $s$ = 0, 1, 2, 3, 4, 5, 6. **(b)** Timing results for varying number of cells $n$ = 100, 150, 250, 600 and fixed number of loci $l$ = 4000. As MEDICC2 was too slow to run on more than 150 cells (with a 2 hour time limit), we exclude it from comparisons where the number of cells $n > 150$.

This provides evidence that even by itself, the ZCNT distance is useful for phylogenetic reconstruction. However, WCND performed at least as well or strictly better than *Lazac-NJ* in terms of median RF and Quartet distance for all but two instances (S3 Fig), emphasizing the importance of accurate estimation of the true evolutionary distance.

## 3.3 Single-cell DNA sequencing data

We used *Lazac* to analyze single-cell whole genome sequencing (WGS) data from 28 human breast and ovarian tumor samples [7]. This dataset was generated using the DLP+ [6], single-cell whole-genome sequencing technology which produces ≈ 0.04× coverage from a median (resp. mean) of 636 (resp. 1457) cells per sample. The original study used Sitka [39], a method that uses the breakpoints between copy number segments as phylogenetic markers, to construct copy number phylogenies using this data.

We found that the phylogenies inferred by *Lazac* are substantially different than the phylogenies constructed by Sitka. Specifically, the normalized RF distance between pairs of phylogenies was greater than 0.5 in most cases, which means that at least half the edges in the phylogeny separate distinct sets of leaves (Fig 3A). The normalized distances were also large across other metrics of tree dissimilarity, including the Quartet and Matching Split metrics. Other summary statistics calculated from the structure of the inferred phylogenetic trees also highlight these differences; for example, while the Sitka phylogenies were highly unresolved, that is, containing many nodes with more than two children, the *Lazac* phylogenies were typically much more resolved.

To further investigate these differences, we first examined whether the more resolved phylogenies inferred by *Lazac* had a meaningful difference in discerning neighboring cells. Specifically, we define the *sibling dissimilarity* metric to be the normalized Hamming distance between the copy number profiles of a pair of siblings in the phylogeny. We found that in 27/28 of the samples, the *Lazac* phylogenies had lower mean sibling dissimilarity scores compared

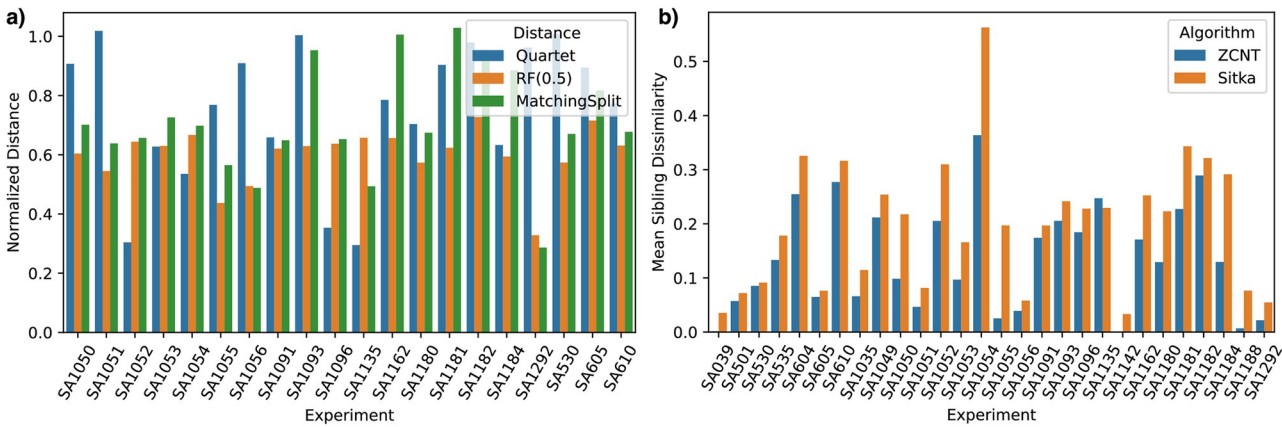

**Fig 3.** **(a)** Normalized distances (Quartet, RF, and Matching Split) between copy number phylogenies inferred by Sitka and *Lazac* on single-cell sequencing of human breast and ovarian tumor samples from [7] with < 1000 cells; **(b)** mean sibling dissimilarity scores for phylogenies inferred from all 28 samples.

to the Sitka phylogenies. In some samples, such as SA1055, the sibling dissimilarity score differs by several orders of magnitude (Figs 3B and 4C). This indicates that the more resolved phylogenies inferred by *Lazac* helps separate cells with distinctive copy number profiles. Or equivalently, the highly unresolved groups of cells in the phylogenies inferred by Sitka contain substantial variation in their copy number profiles.

Second, we analyzed the concordance between the phylogenies and the assignments of cells to copy number clones that was reported in the original publication [7]. The clone labels are obtained from the assignment of each cell to one of $k$ clones, defined by the clustering of cells according to their copy number profiles performed in [7]. For a dataset with $k$ clone labels, the minimum possible clonal discordance score for a tree is $k - 1$, corresponding to the case where each clone label forms a clade in the tree. We find that on 20/28 of the samples, the *Lazac* phylogenies had substantially lower clonal discordance scores than the Sitka phylogenies (S4 and S12 Figs) showing that the *Lazac* phylogenies were more concordant with the copy number clones compared to the published phylogenies.

Third, we verified that *Lazac* better optimizes ZCNT small parsimony score as compared to Sitka. Specifically, we computed the exact ZCNT small parsimony score on the phylogenies inferred by both algorithms using integer linear programming. Unsurprisingly, we found that the *Lazac* inferred phylogenies had lower ZCNT small parsimony scores than Sitka in every case (S9 Fig). Perhaps more surprising, we noticed that on several samples, such as SA1096 and SA1182, the ZCNT small parsimony scores for the Sitka phylogenies were quite close (within 5%) to the scores for the *Lazac* phylogenies. These samples were also the ones where the Sitka phylogenies had lower clonal discordance scores than *Lazac*, supporting the hypothesis that ZCNT parsimony score concords well with the clonal discordance score.

Fourth, we verified that the ancestral labeling inferred by our ZCNT model was biologically meaningful by checking whether it assigned negative copy numbers to ancestors or included amplifications of regions with zero copy number. Negative copy numbers and amplification of zero copy numbers are allowed by the ZCNT model but are violations of the CNT model. In 10/28 patient samples, none of the ancestors were labeled with negative copy number regions and in all 28 samples, none of the ancestors were labeled with a copy number below -1. Copy number of -1 was extremely infrequent: across all samples, the total number of negative stretches normalized by the number of edges in the tree was at most 0.1 (median 0.002) (S10

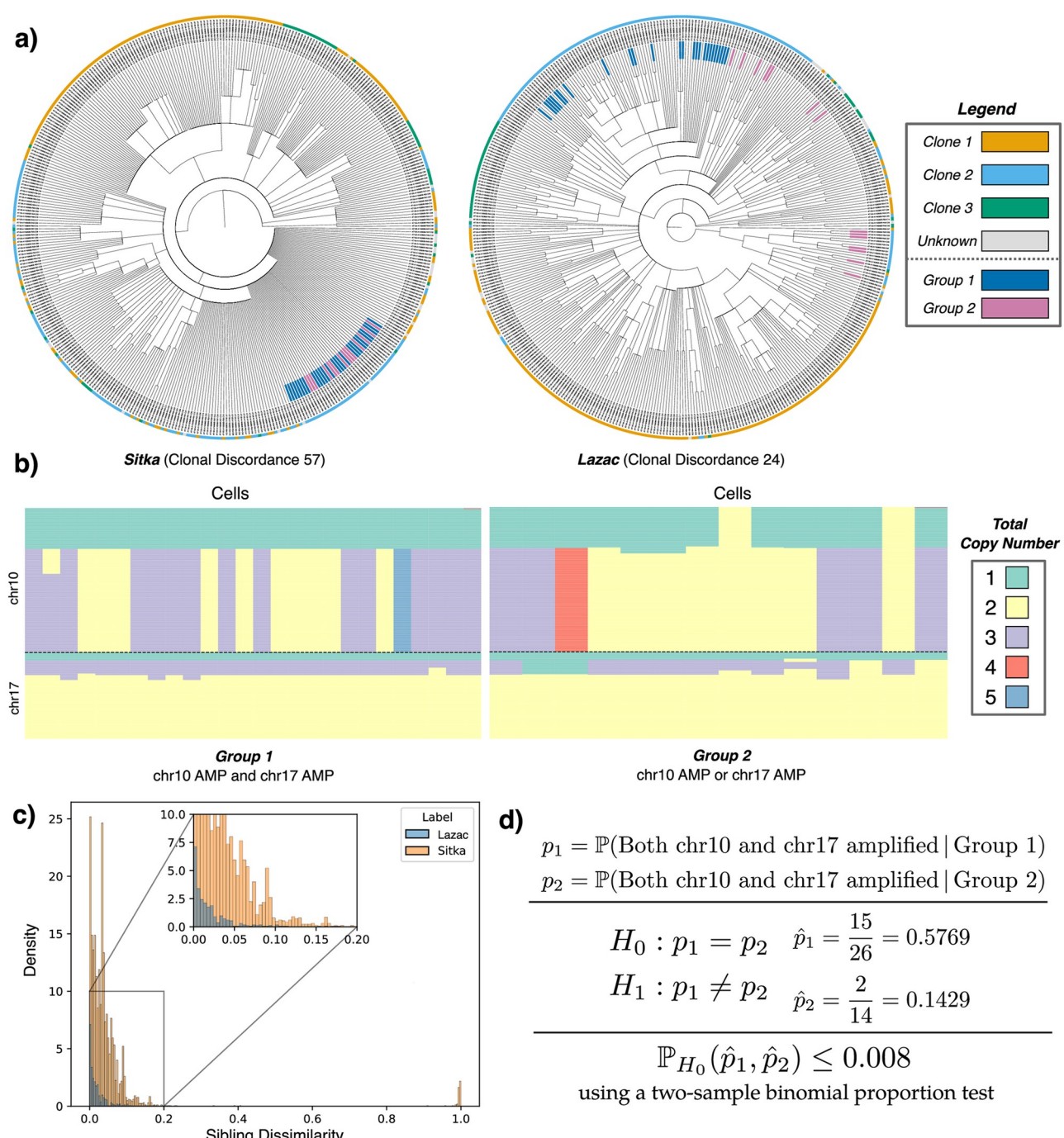

**Fig 4. (a)** The copy number phylogenies inferred by *Lazac* and Sitka on sample SA1292 with the leaves colored by the corresponding clone labels and a subset of cells colored by their assignment to one of two groups (blue and pink). **(b)** The copy number profiles of the cells in (left) Group 1 and (right) Group 2. **(c)** The distribution of sibling dissimilarity scores for sample SA1292. **(d)** The results of a two-sample binomial test of proportion for the observed frequencies of chr10 and chr17 amplifications present in Group 1 and Group 2.

Fig). Amplifications of zero (or negative) copy number loci were also quite rare: across all samples, the total number of such amplifications normalized by the number of edges in the tree was less than 0.33 (median 0.042) (S10 Fig). These CNT violations had negligible effect on the parsimony score: for all samples, CNT violations contributed less than 2% of the ZCNT small parsimony score (S11 Fig). Thus, while our ZCNT model allows for both amplifications of zero copy number loci and negative copy number loci, these events were rare on the patient samples we analyzed and contributed little to the ZCNT small parsimony score.

As a final quantitative evaluation of the *Lazac* and Sitka phylogenies, we examined whether somatic single-nucleotide variants (SNVs) supported the splits in each phylogeny, following the approach of [4]. Note that these SNVs were not used in the inference of either phylogeny, and thus they provide independent validation of the phylogeny. Given the extremely low sequence coverage (0.04× per cell), it is not possible to reliably measure SNVs of individual cells. Thus, we performed this analysis on the three samples (SA039, SA604, SA1035) with the largest number of cells. We identified subtrees in the phylogeny with at least 5% and at most 15% of the cells and identified SNVs present in the subpopulation of cells in these subtrees. Following the approach in [4], we perform a permutation test to determine whether the subtree is supported by more SNVs than expected (Section A.4 in S1 Text). For all three samples, we found that the *Lazac* phylogenies had a greater fraction of supported subtrees ($P < 0.05$) than the *Sitka* phylogenies (S13 Fig). On the largest sample, SA1035, we identified five out of six supported subtrees (supported by 3175, 3334, 3799, 3435, and 3402 SNVs) for the *Lazac* phylogeny compared to only three of eight statistically significant subtrees (supported by 3426, 3129, and 3362 SNVs) for the Sitka phylogeny.

Finally, we performed a qualitative analysis of one of the smaller patient samples, SA1292, to demonstrate the differences between the Sitka and *Lazac* phylogenies. To perform this investigation, we first drew the Sitka and *Lazac* phylogenies for sample SA1292 and colored the leaves by the clone labels (Fig 4A). From this drawing, we noticed that the structure of the Sitka trees was much less resolved (mean degree 6.84) as compared to the *Lazac* trees (mean degree 3.04). We randomly sampled one of these high degree nodes in the Sitka phylogeny, which was the parent of 40 cells. While all 40 cells were siblings in the Sitka phylogeny, in the *Lazac* phylogeny the 40 cells were part of distinct clades. We split the 40 cells into two groups by picking the edge furthest from the root which partitions the cells into two groups, each containing at least 10 cells. We name these groups Group 1 and Group 2, and color the cells from these groups blue and pink, respectively (Fig 4A). We then looked at the copy number profiles of the cells (Fig 4B) and noticed distinct chromosomal aberrations between the two groups. Specifically, 15 of the 26 cells in Group 1 had amplifications in both the *p* arm of chromosome 17 and the *q* arm of chromosome 10. In contrast, only 2 of the 14 cells in Group 2 had both amplifications, a significant difference in proportion ($p < 0.01$; two-sided binomial test of proportion; Fig 4D).

## 3.4 Approximation error of ZCNT small parsimony relaxations

We investigated the approximation error produced by the relaxations (Section 2.4) used in our two polynomial time algorithms for the ZCNT small parsimony problem. To perform this investigation, we first generated a set of 200 copy number phylogenies by perturbing the published phylogeny [39] using from 1 to 200 random nearest neighbor interchange (NNI) operations (Section 3.3). Then, for each phylogeny, we computed the optimal solution to the ZCNT small parsimony problem and its two relaxations using (integer) linear programming.

Importantly, we found that the exact solution to the ZCNT small parsimony problem and the solution obtained by relaxing the integrality condition were nearly identical in every case.

Specifically, they had the exact same solution on 180/200 instances and the solution differed by at most 0.7 on the remaining 20 instances. This leads us to believe that the relaxed linear program often has a special structure, which makes it a good approximation to the ZCNT small parsimony problem.

Removing the balancing condition resulted in solutions with a lower score, implying that the balancing condition does meaningfully constrain the solution space. However, the rankings of phylogenies with and without the balancing condition were highly concordant (Spearman correlation $R^2 = 0.9644$, $p < 10^{-100}$) across the 200 copy number phylogenies (S7 and S8 Figs). Thus, when ranking phylogenies by ZCNT parsimony score—as is done when solving the ZCNT large parsimony problem—removing the balancing condition does not change the result substantially.

## 4 Discussion

We introduced the *zero-agnostic copy number transformation* (ZCNT) model, a relaxation of the CNT model that allows for modification of zero copy number regions. We derived a closed-form expression for the ZCNT distance and presented polynomial time algorithms to solve two natural approximations of the small parsimony problem for copy number profiles. We used our efficient algorithm for the small parsimony problem to derive a method *Lazac*, to solve the large parsimony problem for copy number profiles. We demonstrated that on both real and simulated data, *Lazac* found better copy number phylogenies than existing methods.

There are multiple directions for future work. First, the complexity of the small parsimony problems for both the CNT and ZCNT models remains open, though we conjecture, and provide empirical evidence, that the latter is polynomial. Second, the algorithm we developed for the ZCNT large parsimony problem relies on a simple, hill climbing search using nearest-neighbor interchange operations. We expect that a more advanced approach that uses state-of-the-art techniques from phylogenetics [36, 40] could substantially improve both inference speed and accuracy. Finally, is to apply *Lazac* to other large single-cell WGS datasets [4, 8]. We anticipate that the scalability and accuracy of *Lazac* will be useful in analyzing the increasing amount of single-cell WGS cancer sequencing data.

## Supporting information

**S1 Text.** Supplementary text file (PDF) containing supplementary methods, results, and proofs.
(PDF)

**S1 Data.** Twenty-eight patient copy number phylogenies (Newick) inferred by Sitka and Lazac on single-cell sequencing of human breast and ovarian tumor samples.
(ZIP)

**S1 Table.** CSV file containing the phylogenetic reconstruction accuracy (Quartet and RF) and timing information for CONET simulated data for Lazac, Lazac-NJ, Sitka, WCND, MEDICC2, Hamming-NJ, and Rectilinear-NJ across all simulation parameters.
(CSV)

**S2 Table.** CSV file containing the exact versus relaxed ZCNT small parsimony scores when the balancing and integrality conditions are removed for 200 trees obtained by stochastically perturbing a patient phylogeny SA1053 inferred by Sitka.
(CSV)

**S3 Table.** CSV file containing statistics on the number of children at each node in the inferred *Lazac* and Sitka phylogenies on copy number profiles from 28 human breast and ovarian tumour samples [7].
(CSV)

**S1 Fig.** The exact versus the relaxed score of the optimal solution to the ZCNT small parsimony problem when the balancing condition is removed across 200 phylogenies. The 200 phylogenies were obtained by stochastic perturbation of the phylogeny inferred Sitka [39] on sample SA1053. The dotted line is computed by performing linear regression and is defined by $y = 0.9313 * x + 204.1$ with an $R^2 = 0.972$ and a $p = 1.05 * 10^{-156}$.
(TIFF)

**S2 Fig.** Baseline reconstruction accuracy on (left: RF distance; right: Quartet distance) CONET simulated data across simple NJ methods for copy number tree reconstruction with varying number of cells $n = 100, 150, 200, 250, 600$ across four sets of loci $l = 1000, 2000, 3000, 4000$ and seven random seeds $s = 0, 1, 2, 3, 4, 5, 6$.
(TIFF)

**S3 Fig.** Comparison of reconstruction accuracy (left: RF distance; right: Quartet distance) CONET simulated data across *distance based* methods for copy number tree reconstruction with varying number of cells $n = 100, 150, 200, 250, 600$ across four sets of loci $l = 1000, 2000, 3000, 4000$ and seven random seeds $s = 0, 1, 2, 3, 4, 5, 6$. As MEDICC2 was too slow to run on more than 150 cells, we exclude it from comparisons where the number of cells $n > 150$.
(TIFF)

**S4 Fig.** The relative clonal discordance score $\frac{p_2 - p_1}{p_1 + p_2}$ where $p_1$, $p_2$ are the clonal discordance scores of the *Lazac* and *Sitka* inferred phylogenies respectively.
(TIFF)

**S5 Fig.** Relative difference between the ZCNT distance $d(p, p')$ and the CNT distance computed for patient 8 from a metastatic prostate cancer tumor sample [38].
(TIFF)

**S6 Fig.** Relative difference between the ZCNT distance $d(p, p')$ and the CNT distance for patient 12 from a metastatic prostate cancer tumor sample [38].
(TIFF)

**S7 Fig.** The exact versus the relaxed score of the optimal solution to the ZCNT small parsimony problem when the balancing condition is removed across 200 phylogenies. The 200 phylogenies were obtained by stochastic perturbation of the phylogeny inferred Sitka [39] on sample SA1053. The dotted line is computed by performing linear regression and is defined by $y = 0.9313 * x + 204.1$ with an $R^2 = 0.972$ and a $p = 1.05 * 10^{-156}$.
(TIFF)

**S8 Fig.** The exact and relaxed scores of the optimal solution to the ZCNT small parsimony problem as a function of the number of stochastic perturbations applied to the phylogeny inferred by Sitka [39] on sample SA1053.
(TIFF)

**S9 Fig.** The ZCNT parsimony score for the optimal solution to the ZCNT small parsimony problem for *Lazac* and Sitka inferred phylogenies on copy number profiles from 28 human breast and ovarian tumour samples [7].
(TIFF)

**S10 Fig.** The edge normalized count of zero amplification and negative stretch events in the ancestral labelings of *Lazac* inferred phylogenies on copy number profiles from 28 human breast and ovarian tumour samples [7].
(TIFF)

**S11 Fig.** The ZCNT parsimony score normalized count of zero amplification and negative stretch events in the ancestral labelings of *Lazac* inferred phylogenies on copy number profiles from 28 human breast and ovarian tumour samples [7].
(TIFF)

**S12 Fig.** Comparison of *Lazac* and Sitka clonal discordance scores across 28 human breast and ovarian tumour samples [7].
(TIFF)

**S13 Fig.** Results of somatic SNV analysis on subset of human breast and ovarian tumour samples [7]. Subtrees were called significant if the SNV permutation test p-value was below 0.05 for that clone.
(TIFF)

## Acknowledgments

We thank Uthsav Chitra and Gillian Chu for helpful comments during the preparation of this manuscript.

## Author Contributions

**Conceptualization:** Henri Schmidt, Palash Sashittal, Benjamin J. Raphael.

**Data curation:** Henri Schmidt, Palash Sashittal.

**Formal analysis:** Henri Schmidt, Palash Sashittal, Benjamin J. Raphael.

**Funding acquisition:** Benjamin J. Raphael.

**Investigation:** Henri Schmidt.

**Software:** Henri Schmidt, Palash Sashittal.

**Supervision:** Benjamin J. Raphael.

**Validation:** Henri Schmidt, Palash Sashittal, Benjamin J. Raphael.

**Writing – original draft:** Henri Schmidt.

**Writing – review & editing:** Henri Schmidt, Palash Sashittal, Benjamin J. Raphael.

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
