## [Decision Letter · Decision Letter 0]

20 Sep 2023

Dear Dr. Raphael,

Thank you very much for submitting your manuscript "A zero-agnostic model for copy number evolution in cancer" for consideration at PLOS Computational Biology. As with all papers reviewed by the journal, your manuscript was reviewed by members of the editorial board and by several independent reviewers. The reviewers appreciated the attention to an important topic. Based on the reviews, we are very likely to accept this manuscript for publication, providing that you modify the manuscript according to the reviewers' minor recommendations.

Sincerely,

Philipp M Altrock, Ph.D.

Guest Editor

PLOS Computational Biology

Zhaolei Zhang

Section Editor

PLOS Computational Biology

Reviewer's Responses to Questions

**Comments to the Authors:**

Reviewer #1: Upon reviewing the updated manuscript, I find significant improvements in both its readability and accessibility. The changes introduced have made the content more comprehensible and are likely to appeal to a broader audience. I also appreciate that the authors have convincingly addressed all the concerns raised in the initial feedback.

Reviewer #2: The paper aims to solve an important problem of inferring tumor phylogenies from copy number profiles due to its inherited special property of dependent loci changes. A widely used model for copy number evolution is the CNT model, yet little work has been done to develop methods on CNT model using maximum parsimony. The author introduced a relaxation of the CNT model and provided an efficient approximation algorithm Lazac for the small parsimony, combined with heuristic searching to solve for the large parsimony.

The author tested on the simulation data and showed that the relaxed ZCNT has even the highest accuracy in inferring the copy number phylogenetic tree. The authors also validated the method thoroughly on a real breast cancer single cell dataset and demonstrated that Lazac can output more resolved, biologically meaningful, and consistent results with the original clustering labels, as well as smaller ZCNT score than an existing tool Sitka in the majority of the cases, which also showed a better supports with SNV evidence. The qualitative analysis on sample SA1292 most straightforwardly demonstrated the more reliable result.

The paper is well-written, with clear definitions and comprehensive mathematical proofs for the relaxed version of CNT. Also the parts of comparison with other methods for solving the original CNT as well as the approximation error analysis justifies that the relaxed version has reasonable theoretical foundation as well as little negative influence with regard to actual application.

Since it was previously submitted to RECOMB-CCB, the authors’ solved the previous comments comprehensively and made corresponding edits to the paper.

Comments:

The author used p for both copy number profiles and delta profiles in various definitions and theorems. It might make it more clear if different variable names are used.

Is there any plausible reason for ZCNT performing better than CNT? Could it be some bias in the simulation because it seems a little bit odd to me that ZCNT almost is superior in all the settings.

The paper addresses a critical challenge: inferring tumor phylogenies from copy number profiles, complicated by the inherent property of dependent loci changes. While the CNT model is widely used for copy number evolution, there has been limited development of maximum parsimony methods for it. The author introduces a relaxation of the CNT model and presents the efficient approximation algorithm, Lazac, for small parsimony, complemented by heuristic searching for large parsimony.

The study includes rigorous testing on simulation data, showing that the relaxed ZCNT model consistently exhibits the highest accuracy in inferring copy number phylogenetic trees. Furthermore, the authors conduct a thorough validation on a real breast cancer single-cell dataset, showcasing Lazac's capacity to generate more refined, biologically meaningful, and consistent results, aligned with original clustering labels. Additionally, Lazac consistently produces smaller ZCNT scores compared to the existing tool, Sitka, in most cases, further supported by its better alignment with single-nucleotide variant (SNV) evidence. The qualitative analysis, particularly on sample SA1292, offers a clear demonstration of the method's enhanced reliability.

The paper is well-crafted, with lucid definitions and comprehensive mathematical proofs for the relaxed version of the CNT model. The sections comparing the relaxed version with other methods for solving the original CNT model and analyzing approximation errors provide strong justifications for the theoretical foundation and practical applicability of the approach.

Notably, the authors have diligently addressed previous comments following the paper's earlier submission to RECOMB-CCB, making corresponding edits and enhancing the paper's quality.

Minor comments:

The author used "p" for both copy number profiles and delta profiles in different parts of the paper. It might help make things clearer if they used different variable names.

As for ZCNT with MP performing better than CNT with NJ in all scenarios, is there a plausible explanation for that? Could it be just caused by specific simulation strategy?

**Have the authors made all data and (if applicable) computational code underlying the findings in their manuscript fully available?**

Reviewer #1: Yes

Reviewer #2: Yes

PLOS authors have the option to publish the peer review history of their article (what does this mean?). If published, this will include your full peer review and any attached files.

Reviewer #1: No

Reviewer #2: No

Figure Files:

Data Requirements:

Reproducibility:

References:

---

## [Editor Report · Decision Letter 1]

11 Oct 2023

Dear Dr. Raphael,

We are pleased to inform you that your manuscript 'A zero-agnostic model for copy number evolution in cancer' has been provisionally accepted for publication in PLOS Computational Biology.

Best regards,

Philipp M Altrock, Ph.D.

Guest Editor

PLOS Computational Biology

Zhaolei Zhang

Section Editor

PLOS Computational Biology

---

## [Editor Report · Acceptance letter]

3 Nov 2023

PCOMPBIOL-D-23-01136R1 

A zero-agnostic model for copy number evolution in cancer

Dear Dr Raphael,

I am pleased to inform you that your manuscript has been formally accepted for publication in PLOS Computational Biology. Your manuscript is now with our production department and you will be notified of the publication date in due course.

With kind regards,

Judit Kozma
